# Elevation-Dependent Removal of Cirrus Clouds in Satellite Imagery

**Daniel Schläpfer** [1,*] **, Rudolf Richter** [2] **and Peter Reinartz** [2]

1   ReSe Applications LLC, Langeggweg 3, CH-9500 Wil SG, Switzerland
2   German Aerospace Center (DLR), Earth Observation Center, D-82234 Wessling, Germany;
    rudolf.richter@dlr.de (R.R.); peter.reinartz@dlr.de (P.R.)
*   Correspondence: daniel@rese.ch

**Abstract:**  Masking of cirrus clouds in optical satellite imagery is an important step in automated processing chains. Firstly, it is a prerequisite to a subsequent removal of cirrus effects, and secondly, it affects the atmospheric correction, i.e., aerosol and surface reflectance retrievals. Cirrus clouds can be detected with a narrow bandwidth channel near 1.38 μm and operational detection algorithms have been developed for Landsat-8 and Sentinel-2 images. However, concerning cirrus removal in the case of elevated surfaces, current methods do not separate the ground reflected signal from the cirrus signal in the 1.38 μm channel when performing an atmospheric correction, often resulting in an overcorrection of the cirrus influence. We propose a new operational algorithm using a Digital Elevation Model (DEM) to estimate the surface and cirrus cloud contributions in the 1.38 μm channel and to remove cirrus effects during the surface reflectance retrieval. Due to the highly variable nature of cirrus clouds and terrain conditions, no generic quantitative results could be derived. However, results for typical cases and the achieved improvement in cirrus removal are given for selected scenes and critical issues and limitations of the approach are discussed.

**Keywords:** cirrus detection; cirrus removal; DEM; Landsat-8; Sentinel-2

## 1. Introduction

Satellite imagery is frequently contaminated by high-altitude cirrus clouds in the upper troposphere and in the stratosphere. The occurrence of cirrus clouds is larger than 50% over the midlatitude and tropical regions [1]. Thin cirrus is difficult to detect with visible/near infrared (NIR) bands because land surfaces show a high degree of spatial nonuniformity. Nevertheless, the additive cirrus signal has an adverse effect on the aerosol and surface reflectance retrievals. Therefore, the detection and removal of cirrus, which may strongly vary in a scene, is of particular interest for an improved quantitative processing, and it also helps in the interpretation of data. Hence, it is of scientific interest as well as of practical importance.

Cirrus detection with a narrow bandwidth channel near 1.38 μm was first demonstrated with high-altitude (20 km) AVIRIS data [2], prompting a recommendation to include a dedicated cirrus channel in the MODIS instrument [3]. Later, a cirrus removal algorithm for aircraft and satellite data was published [4–7] and this method of cirrus removal has become a standard process in atmospheric compensation [8]. The 1.38 μm region is characterized by very strong atmospheric water vapor absorption, which blocks the surface reflected solar radiation almost completely but transmits the reflected high altitude cirrus signal.

The frequent occurrence of cirrus clouds and the successful operation of the MODIS cirrus channel are reasons why a dedicated cirrus band has been added to advanced multispectral instruments such as Landsat-8 (L8) [9] and Sentinel-2 (S2) [10]. Additionally, the original cirrus removal method for

visible/near infrared channels in atmospheric window regions was enhanced for S2 type sensors to include a water vapor channel near 0.94 μm [11].

As pointed out by reference [12], cirrus detection with the 1.38 μm band can fail in certain situations (e.g., high elevation, low water vapor content, bright surface) and a narrow band at 1.88 μm is more suitable because of its stronger atmospheric water vapor absorption. However, a 1.88 μm band is currently only implemented in a small number of hyperspectral instruments. The cirrus removal method using multi-angular observations is a further approach which has been successfully applied to GOCI (Geostationary Ocean Color Imager) data [13] but which cannot be applied to single-observation standard satellite imagery.

The availability of cost-free Landsat and Sentinel-2 data and the possibility of processing large time series for monitoring, change detection, and global studies, set new demands on the preprocessing algorithms (e.g., masking of cloud, water, shadow, snow/ice) and surface reflectance retrievals. Among other things it fostered the development of improved automated masking algorithms for Landsat and Sentinel-2 imagery such as the Fmask method [14–16]. Additionally, it fueled the ESA/NASA initiative ACIX (Atmospheric Correction Intercomparison Exercise) [17] to compare various state-of-the-art atmospheric correction processors. This initiative was recently expanded (CMIX—Cloud Masking Intercomparison Exercise) to compare cloud masking algorithms available for L8 and S2 data [18]. Here we present an enhanced cirrus removal algorithm, which uses the per-pixel scene elevation of a DEM to separate the surface and cirrus radiance contributions to the top-of-atmosphere (TOA) radiance in the 1.38 μm channel. It is also an improvement over the approach of [19], because the elevation-dependent cirrus signal was not treated there. This paper presents the main idea and demonstrates the performance with four case studies of L8 and S2 images. The method is fully operational, integrated in the ATCOR (Atmospheric Correction) code [20–23], and applied during the atmospheric correction of L8 and S2 imagery. Table 1 contains the spectral bands and spatial resolution of these sensors.

This paper is organized as follows: Section 2 presents the experimental context of the L8 and S2 sensors and the employed scenes. Section 3 describes the state-of-the-art cirrus masking and removal methods, followed by our proposed method. Section 4 presents results for four selected L8/S2 scenes, including a discussion of the critical issues. The conclusion and possible further improvements are given at the end of the paper.

## 2. Materials

The analysis in this paper is focused on Landsat-8 OLI (level L1T) and Sentinel-2 MS imagery (both S2A and S2B, level L1C) , as the data of these systems is widely used, they offer high quality radiometric measurements and include a dedicated cirrus detection band. Landsat-8 scenes were downloaded from the EarthExplorer (https://earthexplorer.usgs.gov/) and Sentinel-2 scenes were downloaded from Copernicus Open Access Hub (https://scihub.copernicus.eu). Table 1 contains a summary of the spectral bands and spatial resolution of L8 and S2. The surface reflectance retrieval of L8 data is performed without the panchromatic band 8, and prior to the retrieval of S2 data all bands are resampled to a common 20-m grid using bilinear interpolation. The required 1 arcsec Shuttle Radar Topography Mission (SRTM) Digital Elevation Models (DEMs) were downloaded from the USGS web site (https://earthexplorer.usgs.gov/) and in case of S2 resampled to the common 20-m resolution. A total number of 33 scenes from L8 and S2 were analyzed with solar zenith angles ranging from 30 to 60 degrees and ground altitudes between 500 m and 4500 m (compare Appendix Table A1). This case study shows results of four typical scenes from the list.

**Table 1.** Landsat-8 and Sentinel-2 spectral bands and spatial resolution (L8 bands 10, 11: 100 m resolution resampled to 30 m in the L1T product, marked with '*' in list).

| Landsat-8 Bands | Resolution (m) | Sentinel-2 Bands | Resolution (m) |
|---|---|---|---|
| band 1 (0.43–0.45) | 30 | band 1 (0.433–0.453) | 60 |
| band 2 (0.45–0.51) | 30 | band 2 (0.458–0.523) | 10 |
| band 3 (0.53–0.59) | 30 | band 3 (0.543–0.578) | 10 |
| band 4 (0.64–0.67) | 30 | band 4 (0.650–0.680) | 10 |
| band 5 (0.85–0.88) | 30 | band 5 (0.698–0.713) | 20 |
| band 6 (1.57–1.65) | 30 | band 6 (0.733–0.748) | 20 |
| band 7 (2.11–2.29) | 30 | band 7 (0.765–0.785) | 20 |
| band 8 (0.50–0.68) | 15 | band 8 (0.785–0.900) | 10 |
| band 9 (1.36–1.38) | 30 | band 8a(0.855–0.875) | 20 |
| band 10 (10.60–11.19) | 30* | band 9 (0.930–0.950) | 60 |
| band 11 (11.50–12.51) | 30* | band 10 (1.365–1.385) | 60 |
|  |  | band 11 (1.565–1.655) | 20 |
|  |  | band 12 (2.100–2.280) | 20 |

## 3. Method

The proposed method consists of two parts, (i) cirrus detection based on elevation thresholds, and (ii) cirrus removal with the remaining 1.38 µm signal component after subtracting the estimated ground surface contribution.

### 3.1. Part 1: Cirrus Detection

For cirrus detection a useful quantity is the top-of-atmosphere (TOA) reflectance defined as

$$\rho^*(\lambda) = \frac{\pi \, L(\lambda) \, d^2}{E_s(\lambda) \, cos\theta_s} \tag{1}$$

where $\lambda$, $L$, $E_s$, $\theta_s$, and $d$ are wavelength, measured radiance, extraterrestrial solar irradiance, solar zenith angle, and Earth–Sun distance (in Astronomical Units), respectively. The extraterrestrial solar irradiance spectrum $E_s(\lambda)$ is taken from reference [24] and the daily Earth–Sun distance is derived based on an ellipsoidal excentricity of 0.01673.

Cirrus masking is conducted with the $\lambda = 1.38$ µm channel using a certain threshold on the TOA reflectance, e.g., $\rho^*(1.38) > 0.01$, to avoid very thin cirrus or instrument noise. Since atmospheric absorption caused by water vapor is very strong in the 1.38 µm spectral region, the signal corresponding to $\rho^*(1.38)$ is usually exclusively due to cirrus clouds, as the ground reflected solar radiation is absorbed in the lower atmosphere.

However, in the case of very low atmospheric water vapor columns and/or high elevations, a certain fraction of the ground reflected radiation will be included in the TOA reflectance $\rho^*(1.38)$. Therefore, reference [25] recently proposed to employ an empirical elevation-dependent cirrus threshold in the cloud detection of the MAJA (MACCS ATCOR Joint Atmospheric Correction) code:

$$\rho^*(1.38) > (0.007 + 0.007 \cdot h^2), \tag{2}$$

where h is the surface elevation in km above sea level (asl.), taken from a DEM. A pixel is included in the cloud mask if Equation (2) is fulfilled. This inequality is denoted as method "M1" in this paper.

We performed extensive MODTRAN [26,27] simulations to develop a physically-based approach. We calculated the 1.38 µm TOA reflectance as a function of surface reflectance, elevation, MODTRAN5 cirrus models, and water vapor content. When the resulting detection thresholds were applied to S2 data, the radiative transfer based approach of cirrus detection was not successful. The reasons for the failure are probably the difficulty to measure columnar water vapor contents in the presence of cirrus clouds, the limited number of cirrus models in MODTRAN5, the unknown surface reflectance

in the 1.38 µm band (which has to be interpolated with the neighboring bands), and bidirectional reflectance effects in mountainous regions. In addition, since L8 does not have a water vapor band, such an approach could not be applied to L8 data.

Therefore, we concluded that a robust empirical detection threshold function has to be found which performs well for a broad variety of atmospheric situations. First we tested the above criterion (Equation (2)) on the 33 L8 and S2 scenes (Table A1) and found out that it often fails to detect optically thin cirrus ($0.01 < \rho^*(1.38) < 0.02$), especially in elevated areas.

Figure 1 shows examples of the TOA reflectance in the cirrus band at 1.38 µm modeled by MODTRAN5 for a flat 50% reflectance surface and variable ground elevation using the spectral response of the Sentinel-2A instrument. The solar zenith angle was set to 30° and the standard rural aerosol model with 23 km visibility was chosen. The cirrus cloud base altitude was set to 9 km, the top altitude to 10 km, and an extinction coefficient of 0.4 km$^{-1}$ was assigned. The functions with cirrus clouds for the midlatitude summer atmosphere (column water vapor: 2.92 cm) and the midlatitude winter (0.87 cm) standard atmosphere as well as the function for clear sky at midlatitude summer conditions describe the range of typical remote sensing situations. It demonstrates that the water vapor absorption is saturating for ground altitudes up to 1–2.5 km depending on the corresponding water vapor amounts. The impact of ground elevation increases rapidly above 2 km, especially for low water vapor columns. Even though the modeled functions could not be used for cirrus correction directly, they have shown that an empirical correction should consider the ground altitude for elevations greater than about 2 km.

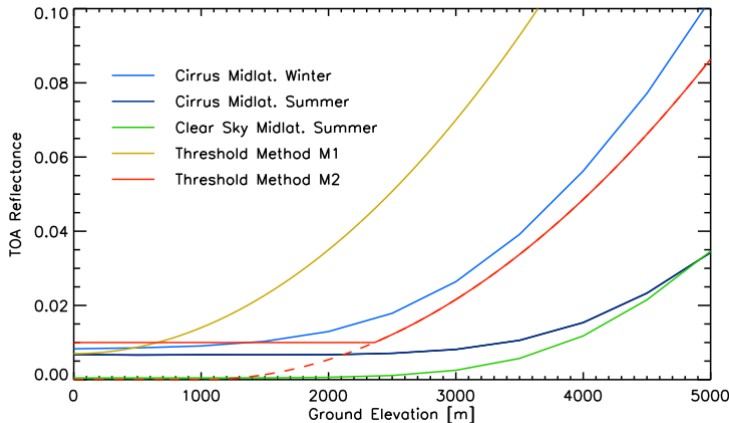

**Figure 1.** Top-of-atmosphere (TOA) reflectance in the 1.38 µm band simulated for Sentinel-2 based on MODTRAN5 runs, compared with the two discussed empirical cirrus detection threshold functions (M1 and M2). The dashed line is the function used for cirrus correction.

As most data acquisitions are not taken from high elevation areas, they typically have medium to high water vapor columns. On the other hand, high altitude mountain regions such as the Alps, Andes, or Himalaya typically have dry atmospheres with low water vapor contents. Therefore, a compromise between these two functions has been sought. Different elevation-dependent functions were evaluated for 33 S2 and L8 scenes (Table A1) and finally a good compromise was obtained with the empirical inequality:

$$\rho^*(1.38) > 0.01, \text{ for h} < 2 \text{ km}, \tag{3}$$

and for altitudes above approximately 2 km, the ground reflectance influence is modeled with a second-order polynomial:

$$\rho^*(1.38) > \left( 0.0054 \cdot (h - 1 \text{ km})^2 \right) > 0.01 \tag{4}$$

This function (last two inequalities), is denoted as method "M2". Figure 1 shows its shape in comparison to typical MODTRAN simulations and compares it to the graph of method M1.

Masking of the scene content is a first necessary step for the aerosol and surface reflectance retrieval. As an example, aerosol retrieval over land uses dark reference pixels [28–30]. Therefore, cloud, shadow, snow/ice, and water have to be excluded during this step in conjunction with the herein described cirrus detection method.

The main purpose of this contribution is the development of an elevation-dependent cirrus removal and atmospheric correction. The cirrus masking methods M1 and M2 serve as templates and the presented cirrus removal algorithm can be adapted to the specific elevation function. Therefore, if a better function is found, it can be used in our approach.

### 3.2. Part 2: Cirrus Removal

The removal of cirrus effects is conducted prior to the aerosol and surface reflectance retrieval, if the scene contained cirrus clouds. The cirrus removal is not conducted with the total signal of the 1.38 μm TOA reflectance but with the signal component remaining after subtracting the estimated ground surface contribution.

Let us denote the above threshold functions of M1 and M2 as $T_{M1}$, and $T_{M2}$, but now with a minimum value of 0.00 instead of 0.01 for M2, i.e., the dashed line in Figure 1. So the total signal $\rho^*(1.38)$ consists of the ground reflected component and the remaining part due to cirrus, i.e.,

$$\rho^*_{cirrus}(1.38 \ \mu m, M1) = \rho^*(1.38 \ \mu m) - T_{M1} \tag{5}$$

$$\rho^*_{cirrus}(1.3 \ 8\mu m, M2) = \rho^*(1.38 \ \mu m) - T_{M2} \tag{6}$$

This means we have to employ the reduced TOA reflectance $\rho^*_{cirrus}(1.38 \ \mu m, M1)$ or $\rho^*_{cirrus}(1.38 \ \mu m, M2)$ during cirrus removal instead of $\rho^*(1.38 \ \mu m)$, which is currently used by standard algorithms.

The standard procedure [5,11,31] uses a scatter plot of the TOA reflectance of dark surface pixels in a reference band versus the 1.38 μm band to obtain a slope coefficient $S(\lambda)$. Then the standard cirrus removal for band $\lambda$ is conducted with $\rho^*(\lambda)$ by subtracting the scaled cirrus TOA reflectance [5,6,11,31].

$$\rho^*_{corrected}(\lambda) = \rho^*(\lambda) - \rho^*(1.38 \ \mu m)/S(\lambda) \tag{7}$$

As mentioned above, it is important to replace Equation (7) with

$$\rho^*_{corrected}(\lambda) = \rho^*(\lambda) - \rho^*_{cirrus}(1.38 \ \mu m, M)/S(\lambda), \tag{8}$$

where M is either M1 or M2. This ensures that only the cirrus signal is subtracted. Before continuing with the aerosol, water vapor, and surface reflectance retrievals, the corrected TOA reflectance is converted into TOA radiance for each band, to be able to run the atmospheric correction with the ATCOR code [20,21]. This code performs a physics-based retrieval using look-up tables calculated with the MODTRAN5 model [26,27].

## 4. Results

This section presents four case studies illustrating the critical issues of cirrus masking and removal of cirrus effects: a Sentinel-2A scene of Morocco, a Landsat-8 scene near Lake Constance, Switzerland, a Sentinel-2A scene from the Rocky Mountains near Albuquerque (NM, USA), and two Sentinel-2A scenes near Railroad Valley (USA). Although the scope of selection is limited due to space reasons and somewhat arbitrary, it nevertheless covers the typical problems encountered when processing cirrus contaminated scenes, i.e., areas with optically thin to very thick (opaque) clouds and altitudes from sea level to about 5 km.

### 4.1. Sentinel-2A Image of Atlas Mountains, Morocco

Figures 2–4 show the results of the first example, a Sentinel-2A scene from Morocco, acquired on 18/01/2016.

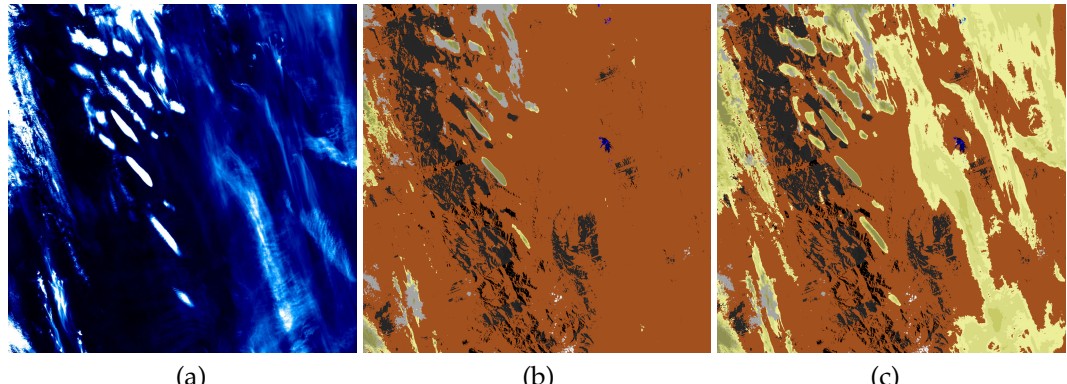

|       |       |       |
| :---: | :---: | :---: |
| (a)   | (b)   | (c)   |

**Figure 2.** Sentinel-2A scene, Morocco, T29RPQ 18/01/2016 (oriented towards east, see also Figure 3), (**a**) cirrus channel, (**b**) M1 cirrus mask, (**c**) M2 cirrus mask. Color coding of masks: orange: cirrus-free, black: shadow, grey: water cloud, blue: water, light-to-darker yellow: increasing cirrus thickness.

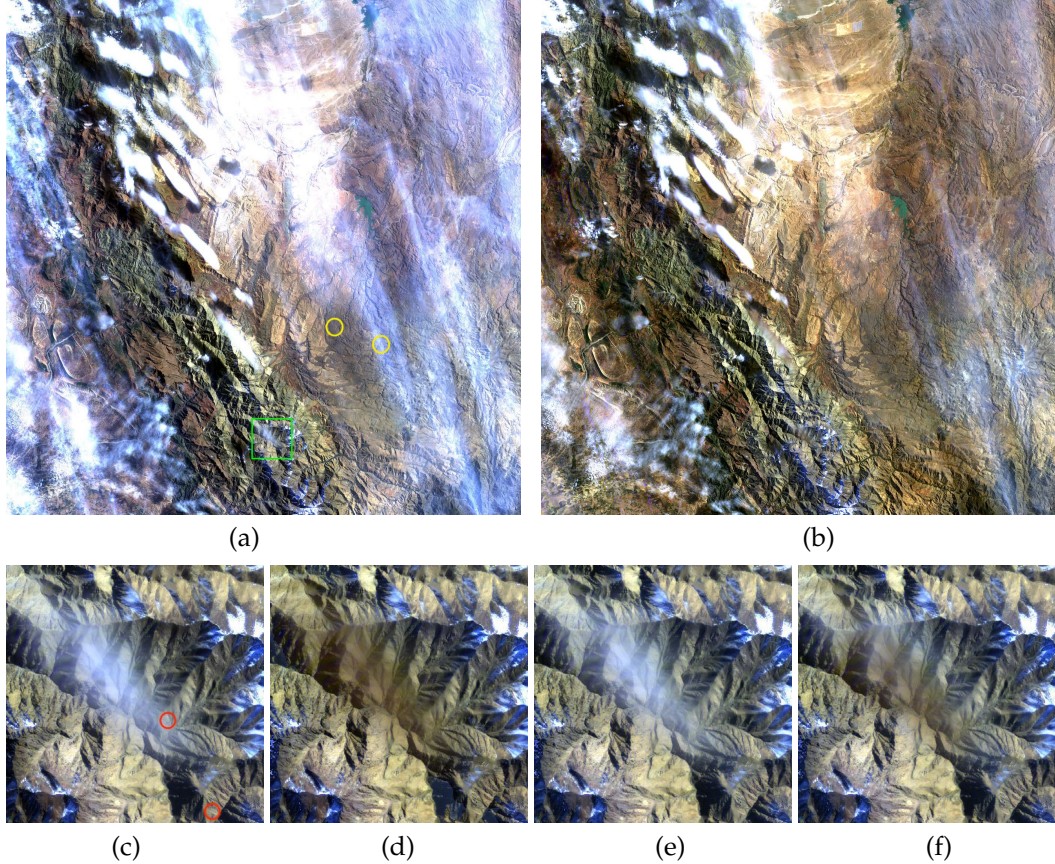

**Figure 3.** Sentinel-2A scene Morocco (oriented towards east), (**a**): original true color radiance image, (**b**): result of cirrus removal with M2. (**c**): subset of green box true color scene after standard atmospheric correction, (**d**) de-cirrus with standard method, (**e**) de-cirrus with method M1, and (**f**) method M2, respectively. The colored circles indicate locations of sample spectra.

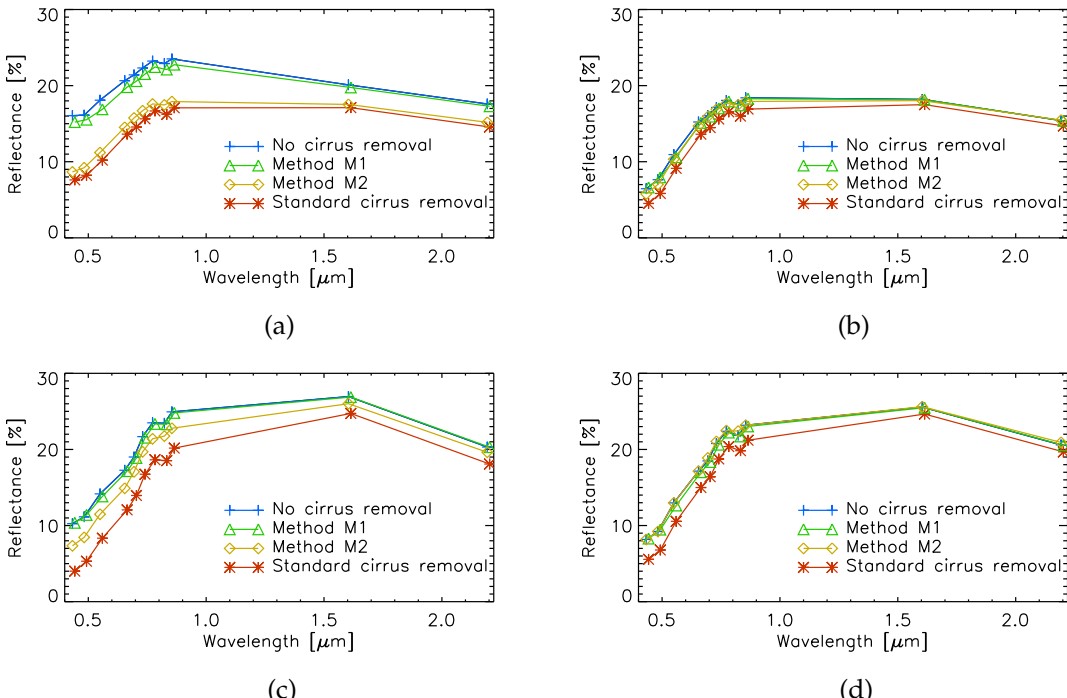

**Figure 4.** (**a**,**b**): spectra from center of yellow circles in Figure 3a; within cirrus (**a**) and outside of cirrus (**b**). (**c**,**d**): spectra from center of red circles indicated in Figure 3c, within thin cirrus area (**c**) and outside of cirrus (**d**).

The solar zenith/azimuth angles for the scene center are 54.9° and 157.8°, respectively. Elevations range between 500 m and 4000 m, and water vapor columns calculated during the atmospheric correction procedure range between 0.5 cm and 2.2 cm. The scene was processed using ATCOR to bottom of atmosphere (apparent) reflectances without correcting the terrain illumination but taking the terrain elevation into account for the path length modelling and the cirrus correction as far as applicable. Figure 2a shows the TOA reflectance image of the 1.38 μm cirrus channel, with maximum values around 0.19 (reflectance units). The southern part of the scene contains thin cirrus with typical reflectance values from 0.01 to 0.04. Figure 2b shows the M1 classification: orange indicates clear areas, shadows are in black, water clouds in grey, and different shades of yellow indicate cirrus. The same color coding is used in the M2 classification in Figure 2c. A comparison of both classification maps with the cirrus channel map shows that the M1 masking misses the majority of the southern cirrus affected areas (DEM between 1500 m and 2500 m) covered by thin cirrus. A closer inspection of the raw image in Figure 3a reveals that these areas are indeed covered by thin cirrus, and some blurred ground features can still be seen.

Figure 3 shows the TOA reflectance data in comparison to the cirrus removal methods (with the estimated cirrus signal component according to Equations (5), (6), and (8)) and a subsequent atmospheric correction. A subset region was selected which contained a small cirrus cloud with $\rho^*_{1.38} < 0.05$ over a ground elevation between 2.5 km and 4.0 km. The variable correction results demonstrate that method M1 underestimates the cirrus signal and does not lead to a viable cirrus correction. Method M2 performs better, whereas the standard method [5,11,31] overcorrects the cirrus contribution resulting in a darker appearance of the area below cirrus cloud.

This statement is also supported by an analysis of the corresponding retrieved surface reflectance spectra. Spectra in Figure 4 are taken as average spectra of 100 pixels each from four areas of the test image. They represent areas of medium to high ground elevations covered by cirrus and neighbouring cirrus-free regions of the same surface type. The correction results for the various methods are shown in comparison to atmospheric compensation without cirrus removal. The first spectra have been taken

in the southern image area (indicated in yellow in Figure 3a) with cirrus signatures of 0.031 and 0.005, respectively, at a ground elevation of approximately 2.0 km.

The results as shown in Figure 4a,b are comparable between the standard method and method M2, i.e., both methods correct the cirrus effect, while method M1 does not detect and correct the cirrus effect. A second pair of samples was taken within the selected subset for a situation at high ground elevations (about 2.8 km). Two spectral samples were taken from the image (indicated by red circles in Figure 3c) with a cirrus signature of $\rho^*_{1.38} = 0.036$ and $\rho^*_{1.38} = 0.010$, respectively. The corresponding correction results are shown in Figure 4c,d. Here, the standard method leads to a lower reflectance spectrum than the average of the reference area, whereas method M2 is closer to the expected reflectance values. This confirms the overcorrection effect at high ground elevations.

### 4.2. Landsat-8 Image of Lake Constance, Switzerland

The next example presents a small subset of a Landsat-8 scene containing Lake Constance and its rural environment in Switzerland. The acquisition date is 19/07/2014 and the solar zenith/azimuth angles are 30.8° and 142.5°, respectively. For reasons of space, we do not include images of the whole scene but select a smaller region adjoining Lake Constance. Figure 5a shows a small section of this lake in the north-eastern part of the image. The scene exhibits thin cirrus clouds in the center, which can clearly be seen in the 1.38 µm TOA reflectance image (Figure 5b).

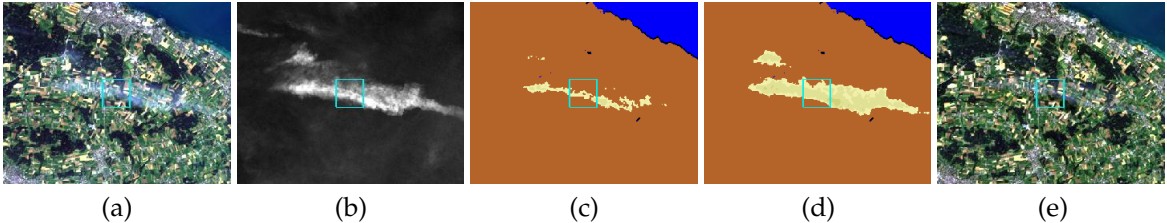

|     (a)     |     (b)     |     (c)     |     (d)     |     (e)     |

**Figure 5.** Landsat-8 subset, Lake Constance, 19/07/2014, (**a**) true color, (**b**) cirrus channel, (**c**) M1 cirrus mask, (**d**) M2 mask (same color coding as in Figure 2), (**e**) surface reflectance product.

The elevations of the subset vary between 385 m (asl.) in the lake region and a maximum of about 700 m in the south-west scene. The cyan box in the center of the scene marks the central location of the cirrus clouds with an average elevation of 524 m. Figure 5 shows the M1, M2 cirrus classification maps and a comparison with the 1.38 µm TOA reflectance map indicates that method M1 does not detect thin cirrus. Figure 5e is the level 2 (L2) product, i.e., surface reflectance (RGB = 650, 560, 443 nm) after the M2 cirrus removal. Method M1 yields a result that is visually very similar to M2 and is not displayed. Compared to the uncorrected product, the effects of thin cirrus clouds were successfully removed.

Figure 6 shows the comparison of averaged surface reflectance spectra of M1 and M2. The left graphs (Figure 6a,c) compare the M1 and M2 reflectance spectra with the atmospheric correction result without cirrus removal for the cirrus area using an average of all 9700 pixels in the M2 cirrus mask of Figure 5d. M1 values are slightly higher because of the smaller estimate of the cirrus contribution. The right graphs (Figure 6b,d) presents a statistical comparison of a mask of the same size in clear sky conditions, in an area southernly adjacent to the first mask. Only minor differences in retrieved surface reflectance spectra can be found for this area. If comparing the average spectra of the cirrus region to the average spectra of the corresponding adjacent cirrus-free area, M2 reflectance results agree within approximately 10% relative accuracy, while M1 deviations are significantly higher at up to 20%.

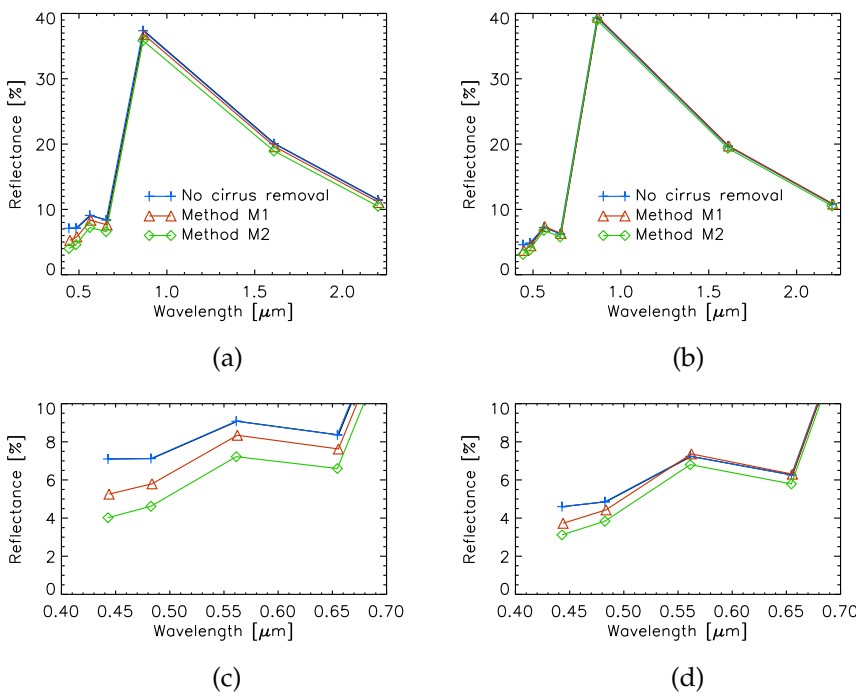

**Figure 6.** Surface reflectance spectra from cirrus region of Figure 5 (**a**,**c**) and from adjacent clear sky region (**b**,**d**) with no significant cirrus influence, using M1 and M2, respectively. Graphs (**c**,**d**): same spectra as (**a**,**b**), enlarged for the visible part of the spectrum.

### 4.3. Sentinel-2A Image of Rocky Mountains Near Albuquerque, USA

The third example deals with a S2 scene of the Rocky Mountains near Albuqerque containing very low atmospheric water vapor. The acquisition date is 30/07/2017 and the solar zenith/azimuth angles are 59.2° and 163.1°, respectively. Again, for reasons of space we do not show the complete scene, but a smaller subset, with elevations ranging from 1800 m to 3000 m asl. Figure 7a–d present the original true color image, the 1.38 µm TOA reflectance ranging between 0 and 0.02, the classification map using a standard threshold of 0.01, and the new cirrus mask which is the same for methods M1 and M2. The TOA reflectance map of the 1.38 µm channel clearly exhibits the mountain structure and its surface, because of the extremely low water vapor content ranging from 0.1 cm to 0.3 cm calculated from the S2 scene during atmospheric correction.

A comparison with the DEM map shows that the very high mountain regions (2700–3000 m asl., coded white) in the center usually correspond to higher values of the 1.38 µm TOA reflectance map. The M1 and M2 maps are identical due to relatively low cirrus TOA reflectance (<0.026) and high elevations, i.e., if the threshold $T_{M1}$ or $T_{M2}$ of the TOA reflectance exceeds the measured $\rho^*(1.38)$, then the cirrus contribution $\rho^*(1.38, cirrus)$ is set to zero. Thus, the terrain dependent functions M1 and M2 both avoid false detection of cirrus clouds and no unnecessary cirrus correction is done for this sample scene.

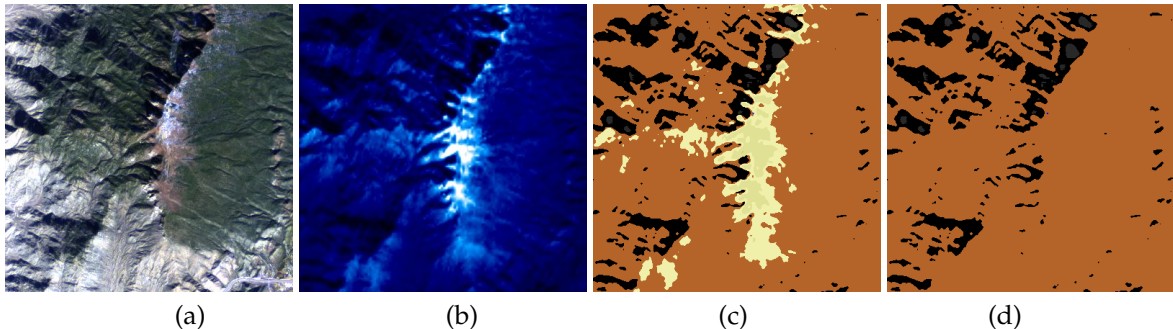

**Figure 7.** Sentinel-2 subset, Rocky Mountains near Albuquerque, 06/12/2015, (**a**) true color, (**b**) cirrus channel, (**c**) standard cirrus mask, (**d**) identical M1, M2 masks with no cirrus detected.

### 4.4. Sentinel-2A Image of Railroad Valley, USA

The last example illustrates two Sentinel-2A scenes from the Railroad Valley area. We tried to find a cirrus-affected and a cirrus-free scene acquired within three days and close to nadir observation. However, it is not an easy task to find a completely cirrus-free counterpart for a cirrus contaminated scene. Finally, we selected the two scenes of 30/07/2017 (solar zenith angle 25.6°, azimuth angle 136.2°), and 02/08/2017 (solar zenith angle 25.0°, azimuth angle 141.9°). Both are partly cirrus-contaminated but it is possible to select cirrus-free regions in the 30/07/2017 scene that are covered by cirrus in the 02/08/2017 image.

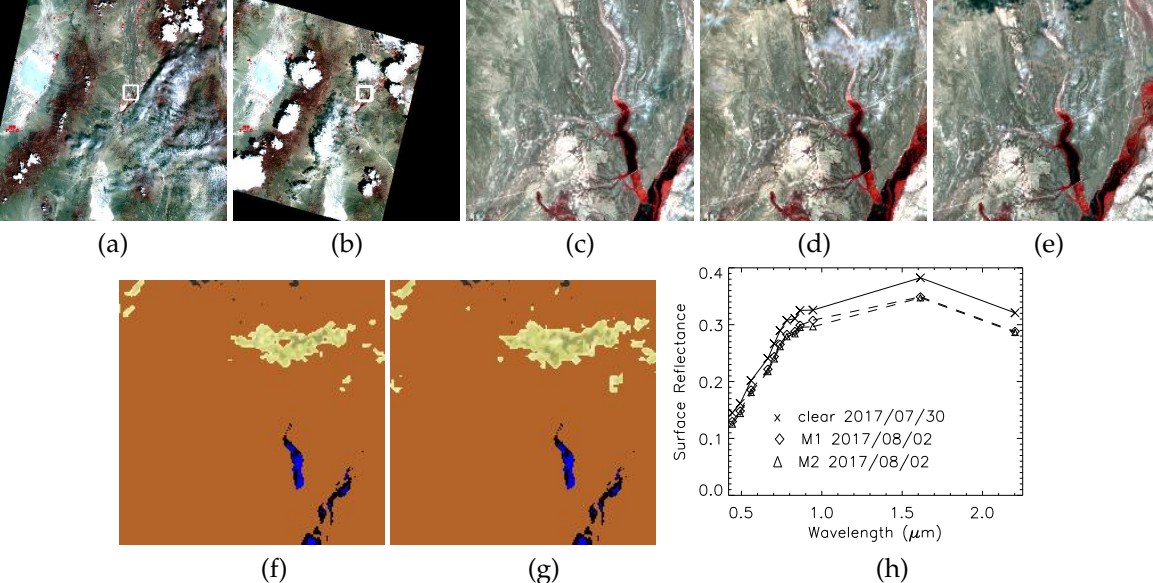

**Figure 8.** Sentinel-2 Railroad Valley, CIR rendition, (**a**) 30/07/2017; (**b**) 02/08/2017; (**c**) white box subset of (**a**); (**d**) white box subset of (**b**); (**e**) cirrus removal with method M2 (visually very similar to M1); (**f**) subset M1 cirrus mask; (**g**) M2 cirrus mask (same color coding as in Figure 2); (**h**) subset spectra of clear part (30/07/2017) and M1, M2 spectra from cirrus-affected scene 02/08/2017.

The selected scenes differ less than 1° in the solar zenith angle (25.6° and 25.0°) and less than 6° in the view azimuth angle, and both viewing angles are close to nadir (< 5°). The selected area is flat with an elevation of 1440 m above sea level. The results are presented in Figure 8: top left to right show color infrared (CIR) renditions of the scenes (a) 30/07/2017, (b) 02/08/2017, (c) a cirrus-free region of 30/07/2017 marked with a white box in the full scene, (d) the corresponding cirrus-affected region of 02/08/2017, and (e) the cirrus removal of (d) with method M2.

Method M1 yields very similar results as M2, and there is no visual difference. This is caused by the similar cirrus masks of M1 and M2 presented in Figure 8f,g. A quantitative spectral comparison in terms of surface reflectance spectra is given in Figure 8h. The M1 M2 spectra are based on the statistics of 3700 pixels from the M2 cirrus mask and represent the average over these pixels. The difference in standard deviation is less than 0.0015 reflectance units for all bands. Both spectra retrieved from the contaminated scene closely agree. The shape of the retrieved spectra is very similar to the corresponding spectrum from the cirrus-free region. Reflectance differences between the retrieved spectrum (M1, M2) and the corresponding cirrus-free spectrum range between 0.015 to 0.025 (reflectance units) in the VNIR and 0.03 in the SWIR, with a very similar spectra. This difference may be attributed to the presence of clouds adjacent to the measurement area (compare Figure 8a) for the cirrus-affected subset, which may have reduced the irradiance field in a way which cannot be modelled in the physical atmospheric compensation.

## 5. Discussion

The presented semi-empirical method was developed in an iterative way based on the listed 33 test scenes and in conjunction with MODTRAN based simulations. Validation was done on samples, whereas rigorous validation of the results of cirrus removal would require a broad collection of known ground truth reflectance spectra. The availability of such a database is very unrealistic, considering the logistic investment, time and cost, especially in remote high mountain areas. In addition, a few experiments would not be sufficient to obtain a statistically sound database. Under these circumstances a practical approximation as offered by our method is a reasonable alternative.

At this point a note of caution is required: in the ideal case a cirrus abundance map should be calculated, since there is no simple on/off for a cirrus mask. Instead, cirrus optical thickness and its corresponding 1.38 μm TOA reflectance can take continuous values from very small to large. In practice, this abundance map cannot be calculated from a single cirrus channel, because there are several unknowns; the 1.38 μm TOA reflectance depends on:

- Surface albedo and surface bidirectional reflectance distribution function (BRDF), especially for sloped mountain regions,
- Cirrus optical thickness and cirrus particle size,
- Aerosol content and haze below cirrus cloud, and
- Atmospheric water vapor.

One might consider to interpolate the unknown 1.38 μm surface reflectance after atmospheric correction from neighboring spectral bands, but in the case of L8 and S2 the corresponding bands in the NIR and short-wave infrared are not close enough for an accurate interpolation (see Table 1). Although atmospheric water vapor is a major influencing factor, it cannot be considered quantitatively in the cirrus correction. Landsat-8 is not equipped with a water vapor band, so there is no possibility to introduce a water vapor correction factor for L8 data. Even for S2 data a water vapor correction factor to Equations (4)–(8) is not possible, because the cirrus affected parts of a scene also influence the TOA radiance in the NIR channels used for water vapor retrieval (865 nm and 945 nm in case of S2) causing an inaccurate value for the water vapor column. Since no quantitative relationship between water vapor content and cirrus TOA reflectance can be derived from S2 scenes, a compromise is needed to achieve robust results for the removal of cirrus effects in optical imagery. Our proposed Equations (4)–(8) yields satisfying results using the continuous signal of the TOA cirrus reflectance as input.

A second cross-sensitivity observed during the evaluation is the aerosol detection and correction. Depending on the cirrus abundance, the automatic aerosol detection routines based on dark dense vegetation approaches will lead to variable results which may add errors in the reflectance retrieval. Minor variations in the presented spectra can mostly be attributed to this effect.

## 6. Conclusions

A novel method of elevation-dependent cirrus removal is presented, which separates the ground reflected signal from the cirrus signal in the 1.38µm channel. The preprocessing step of cirrus masking is performed by an elevation-depending function, and results are shown for two template functions (M1, M2). Function M1 employs a conservative mask, which omits thin cirrus clouds, while function M2 additionally masks thin cirrus. Both functions are a compromise, since M1 yields a larger omission error of cirrus classification, while M2 has a higher commission error, especially in cases of high mountain areas (>2 km) and dry conditions. Both template functions yield better results in high elevation regions (>2 km) than the standard method of cirrus removal which neglects the elevation influence, thereby often overcorrecting for cirrus.

No general quantitative figure of the improvement with respect to the current standard method can be given since the results strongly depend on the cirrus scene content and parameters that cannot be retrieved from L8 and S2 images. Nevertheless, the presented case studies show an improvement in terms of visual quality and in surface reflectance spectra quality. The proposed method is suitable for operational processing chains and has been included in the ATCOR code.

Other cirrus cloud detection methods are using the image statistics including thermal band data, if available. Cirrus cloud masking results with the Fmask model strongly depend on the user-supplied cloud probability threshold, which offers the possibility to optimize the processing per scene iteratively but makes it difficult to select the best compromise for scenes all over the world.

This contribution is also intended as a note of caution concerning the masking and processing of cirrus affected imagery. Further research is needed to overcome the limitations of current operational methods for cirrus contaminated scenes and to improve the cirrus masking function. A promising way could be through the use of contextual image information, the exploitation of image blurring, and the fuzzy appearance of cirrus clouds.

**Author Contributions:** D.S.: methodology, validation, software, writing of article; R.R.: concept, methodology, software, writing of article; P.R.: validation, writing of article. All authors have read and agreed to the published version of the manuscript.

**Funding:** This research received no external funding.

**Conflicts of Interest:** The authors declare no conflict of interest.

## Appendix A

**Table A1.** Landsat-8 level L1T and Sentinel-2 level L1C test scenes.

| Scene/Sensor | Location | Country | Date (DD/MM/YYYY) | Path/Row (L8), Tile (S2) |
|---|---|---|---|---|
| 1 / S2 | Atlas Mountains | Morocco | 18/01/2016 | T29RPQ |
| 2 / L8 | Lake Constance | Switzerland | 19/07/2014 | 194/27 |
| 3 / S2 | Railroad Valley | USA, NV | 30/07/2017 | T11SPC |
| 4 / S2 | Railroad Valley | USA, NV | 02/08/2017 | T11SPC |
| 5 / S2 | Klamath N.Forest | USA, CA | 25/09/2016 | T10TDL |
| 6 / S2 | La Paz | Bolivia | 18/03/2016 | T19KEB |
| 7 / S2 | San Francisco | USA, CA | 17/11/2015 | T10SEG |
| 8 / S2 | Pyrenees | Spain | 05/03/2015 | 197/31 |
| 9 / L8 | Kathmandu | Nepal | 24/01/2015 | 141/14 |
| 10 / L8 | Lahore | Pakistan | 10/11/2013 | 149/38 |
| 11 / L8 | Kitzbühl, Alps | Austria | 17/10/2017 | 192/27 |
| 12 / L8 | Quebec | Canada | 24/05/2015 | 12/26 |
| 13 / S2 | Diyarbakr | Turkey | 20/04/2016 | T37SFC |
| 14 / S2 | St. Johann, Alps | Austria | 09/05/2016 | T32TQT |
| 15 / S2 | Davos, Alps | Switzerland | 22/05/2016 | T32TNS |
| 16 / L8 | Beijing | China | 13/04/2014 | 123/32 |
| 17 / S2 | Salon-de-Provence | France | 15/02/2016 | T31TFJ |

**Table A1.** *Cont.*

| Scene / Sensor | Location | Country | Date (DD/MM/YYYY) | Path/Row (L8), Tile (S2) |
|---|---|---|---|---|
| 18 / L8 | Montelimar | France | 14/07/2013 | 196/29 |
| 19 / L8 | Rio de Janeiro | Brazil | 18/06/2014 | 217/76 |
| 20 / L8 | Sitten | Switzerland | 30/08/2015 | 195/28 |
| 21 / L8 | Basel | Switzerland | 25/09/2013 | 195/27 |
| 22 / L8 | Orurillo | Peru | 07/02/2017 | 3/70 |
| 23 / L8 | Murnau | Germany | 01/09/2015 | 193/27 |
| 24 / S2 | Gobabeb | Namibia | 06/03/2019 | T33KWP |
| 25 / S2 | Madrid | Spain | 15/11/2017 | T30TVK |
| 26 / S2 | Munich | Germany | 16/06/2017 | T32UPU |
| 27 / L8 | Munich | Germany | 01/09/2015 | 193/27 |
| 28 / S2 | Railroad Valley | USA, NV | 23/07/2017 | T11SPC |
| 29 / S2 | Blair | USA, NE | 16/11/2019 | T15TTG |
| 30 / L8 | Darjeeling | India | 15/03/2015 | 139/41 |
| 31 / S2 | Seattle | USA, WA | 06/04/2019 | T10TET |
| 32 / S2 | Barrax | Spain | 11/10/2019 | T30SWJ |
| 33 / S2 | Barrax | Spain | 20/11/2019 | T30SWH |

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
