# Peer review of "Elevation-Dependent Removal of Cirrus Clouds in Satellite Imagery"

_remotesensing, doi:10.3390/rs12030494_

Round 1
Reviewer 1 Report
interesting article on the improvement of an existing method, well written and clearly presented.
REMARK 1: (no update is required, just a suggestion)
Figure 1 : I can see that the detection / correction curve for M2 is created on the basis of MODTRAN5 simulations and comparison with multiple images. It would be interesting and make the atricle more complete if more information was given on the way this curve was created : what MODTRAN settings are applied to produced this curve (especially on cirrus model) or is it some kind of average of many curves? what kind of response curve did you assume for the 1.38µm simulations ? etc...
REMARK 2: (update required)
at a certain point, the article refers to two cirrus detection methods M1 and M2. At first read, it was a bit difficult to know what exactly was M1 and M2. I think it is interesting to add, after the de description of a method, an annotation like "Further referred to as M1" or something similar
REMARK 3: (update required)
Figure 2. has to be updated : lacks one sub-image, the true-color one, i think.
REMARK 4 : (cosmetic suggestion, update not necessary)
figure 6 : the spectra are plotted incompletely. the nir band falls outside the plot (probably due to vegetation content). if this is for clarity reasons, you can remove the band (and mention this somewhere) from the plot, so you get a full curve. Otherwise, please include this band.
Reviewer 2 Report
Comments and Suggestions for Authors:
This paper proposes the removal cirrus clouds for the atmospheric correction. As the authors mention in this research, it is very important for the atmospheric correction to detect and remove clouds, especially cirrus clouds. If the following minor revisions are accepted for authors, it is more understandable for readers, hopefully.
Comments as follow:
L64-73
- It is better to show the sensor name, OLI and MSI clearly.
- Are L8 OLI and S2 MSI data downloaded from EarthExplorer ?
- Did you use both of S2A and S2B ?
- Did you use L8 OLI L1T and S2 MSI L1C ?
L79
- Which irradiance model did you use ?
- Could you show the equation and a reference for Earth-Sun distance ?
3. L120
If the scene contains cirrus clouds the removal of cirrus effects is conducted prior to the aerosol and surface reflectance retrieval.
->
If the scene contains cirrus clouds, the removal of cirrus effects is conducted prior to the aerosol and surface reflectance retrieval.
4. L139
4.1 Sentinel-2A Atlas Mountains, Morocco
->
4.1 Sentinel-2A image of Atlas Mountains, Morocco
5. L178
4.2 Landsat-8 Lake Constance

 ->
4.2 Landsat-8 image of Lake Constance
6. L181
For space reasons we do no include images of the whole scene,
->
For reasons of space, we do not show the whole L8 imagery.

7. L185
The elevations of the subset between 385 m above sea level (asl) in the lake region, the land average is around 530 m, with a maximum of about 700 m in the south-west scene.
-> (This sentence is not clear.)
8. L187
Fig.’s -> Fig.
9. L.193
Surface reflectance spectra with methods M1 and M2 are compared, see Fig. 6.
->
Fig. 6 shows the comparison of surface reflectance spectra with methods M1 and M2.
10. L193
The left graphic compares the M1 and M2 spectra to the atmospheric correction result ɉ۬
->
The left graphic compares between the M1 and M2 spectra to the atmospheric correction result …
11. L200
- within ∼ 10% -> within 10%
- up to ∼ 20% -> up to 20%

12. L201
4.3. Sentinel-2 Rocky Mountains near Albuquerque

 ->
4.3. Sentinel-2A (or Sentinel-2B ??) image of Rocky Mountains near Albuquerque
13. L204
for space reasons we do not show the complete scene,
->
for reasons of space, we do not show the complete scene,
14. L214
if the surface estimated TOA reflectance …

 ->
if the surface reflectance estimated TOA reflectance …
15. L218
4.4. Sentinel-2 Railroad Valley, 2017/07/30 and 2017/08/02
->
4.4. Sentinel-2A images of Railroad Valley on 2017/07/30 and 2017/08/02
